# Non-Invasive Methods for Assessing the Welfare of Farmed White-Leg Shrimp (*Penaeus vannamei*)

**DOI:** 10.3390/ani13050807

**Published:** 2023-02-23

**Authors:** Ana Silvia Pedrazzani, Nathieli Cozer, Murilo Henrique Quintiliano, Camila Prestes dos Santos Tavares, Ubiratã de Assis Teixeira da Silva, Antonio Ostrensky

**Affiliations:** 1Wai Ora—Aquaculture and Environmental Technology Ltd., Curitiba 80240-050, Brazil; 2Graduate Program in Animal Science, Federal University of Paraná, Curitiba 80035-050, Brazil; 3Integrated Group of Aquaculture and Environmental Studies (GIA), Animal Science Department, Agricultural Sciences Sector, Federal University of Paraná, Curitiba 80035-050, Brazil; 4FAI Farms, Londrina 86115-000, Brazil; 5Graduate Program in Zoology, Federal University of Paraná, Curitiba 81531-980, Brazil

**Keywords:** aquaculture, grow-out pond, larval rearing, penaeid, shrimp farming, well-being

## Abstract

**Simple Summary:**

Each year, approximately 167 billion *Penaeus vannamei* (white-leg shrimp) are farmed worldwide from an estimated total of more than 400 billion marines and freshwater shrimp farmed. In this context, the welfare of decapod crustaceans, the group with the most farmed animals on the planet, is becoming an increasingly important issue for researchers and society, and this debate will soon reach shrimp labs and farms. This article presents protocols specifically designed to measure the welfare of *P. vannamei* at all stages of their production cycle, from reproduction through larval rearing and postlarval transport to juvenile rearing in earthen ponds. These protocols were developed using four domains of welfare: nutrition, environment, health, and behaviour. Together, they help assess the fifth domain: psychology. The assessment protocols also include reference values for each indicator and three possible values for animal welfare on a continuum from positive (score 1) to very negative (score 3). Our assessment protocols can identify the critical points in the shrimp aquaculture process and are an essential step towards improving the welfare of farmed shrimp worldwide.

**Abstract:**

Gradually, concern for the welfare of aquatic invertebrates produced on a commercial/industrial scale is crossing the boundaries of science and becoming a demand of other societal actors. The objective of this paper is to propose protocols for assessing the *Penaeus vannamei* welfare during the stages of reproduction, larval rearing, transport, and growing-out in earthen ponds and to discuss, based on a literature review, the processes and perspectives associated with the development and application of on-farm shrimp welfare protocols. Protocols were developed based on four of the five domains of animal welfare: nutrition, environment, health, and behaviour. The indicators related to the psychology domain were not considered a separate category, and the other proposed indicators indirectly assessed this domain. For each indicator, the corresponding reference values were defined based on literature and field experience, apart from the three possible scores related to animal experience on a continuum from positive (score 1) to very negative (score 3). It is very likely that non-invasive methods for measuring the farmed shrimp welfare, such as those proposed here, will become a standard tool for farms and laboratories and that it will become increasingly challenging to produce shrimp without considering their welfare throughout the production cycle.

## 1. Introduction

### 1.1. Animal Welfare and Progress in Shrimp Farming

Agricultural, aquacultural, and industrial production chains only develop (and grow) because they change. The shrimp farming production chain is considered one of the world’s most controversial agri-food systems for animal protein production [1,2,3,4,5]. The evolution of shrimp farming practices is currently taking place on several fronts. For example, there is a trend towards intensification of production systems for better utilisation of resources [6,7,8,9,10]; the pursuit of certifications and regulations to adapt shrimp production to the new demands of the market and society in general [11,12,13]; genetic improvement of animals [12,14,15]; improvement of shrimp feeding and nutrition [16,17]; and increased efforts towards hygienic-sanitary controls and biosecurity in shrimp farms [18,19], to name a few. This scenario of change, based on the scientific and technological development of the sector, in turn, helps to understand why the global production of a single species, the white-leg shrimp *Penaeus vannamei*, has increased by almost 53% in 5 years, from 3803.6 thousand tonnes to 5812.2 thousand tonnes between 2015 and 2020, so that this species already accounts for 51.7% of total shrimp production [20]. It is estimated that more than 167 billion shrimps of this species are produced annually [21] and that the total number of shrimps and prawns farmed annually reaches about 440 billion [22].

However, it is human nature that despite all this progress, there is often a temptation to cling to traditional production methods and concepts that seemed to work satisfactorily in the past, even though they certainly no longer work in the same way or are no longer acceptable under present and future conditions. Perhaps this is a challenge to overcome regarding welfare in shrimp farming.

The term “welfare” describes a potentially measurable quality of an animal’s life at a particular point in time and is, therefore, a scientific concept [23]. However, this concept is fundamentally dynamic and related to the adaptive abilities of the individual [24]. In other words, the welfare of a particular animal is always a temporary state related to the animal’s experiences and moves on a continuum from negative to positive [25]. Thus, to improve animal welfare, it is necessary to ensure that the animal’s experience is as positive as possible. Although society has recognised the importance of animal welfare in the commercial production of homeothermic vertebrates [26,27,28,29] and even ectotherms [30,31,32], there is still a glaring lack of recognition of the welfare of higher invertebrates, as in the case of shrimps [33,34,35].

The discussion on the need to apply the principles of animal welfare to shrimp farming includes two main aspects: those related to the current scientific evidence for sentience in these animals [36,37,38] and the ethical/legal aspects associated with the issue [39,40,41].

### 1.2. Shrimp Sentience

From the first perspective, the welfare of an organism is inseparable from the degree of suffering and the positive states that this individual experiences at a certain point in time [42]. Sentience, thus, refers to the ability of an animal to consciously perceive what is happening to it and what surrounds it, consciously perceive through the senses, and consciously feel or subjectively experience [43]. Crump et al. [44] explained that sentience needs to be understood and addressed in a broad sense, including sensory experiences (visual, auditory, tactile, and olfactory) as well as feelings of warmth or cold, fatigue, hunger, thirst, boredom, excitement, fear, pain, pleasure, and joy. These authors also pointed out that this ability to feel must be distinguished from other related abilities, since a sentient being, for example, is not necessarily able to think about or understand the feelings of other animals. Finally, they argue that in answering questions about the ability of invertebrate animals to feel, we must rely (at least partly) on behavioural and cognitive markers associated with knowledge of their nervous systems.

According to Birch et al. [45] and Feinberg and Mallatt [46], invertebrates’ brains differ even more radically from those of mammals and fish, the vertebrates most distant from humans. These authors explain that, although there are more than 500 million years of evolution between invertebrates and humans, it cannot be assumed that invertebrates lack sentience simply because their cerebral ganglia are organised differently from the brain of vertebrates [45,46]. In the higher invertebrates (chordates, echinoderms, arthropods, and mollusks) there are a variety of complex intra- and interspecific neuronal structures that determine the degree of their sentience [47]. According to Conte et al. [48], decapod crustaceans have sensory cells that enable the perception of noxious stimuli and show behavioural responses to these stimuli. However, the authors added that these responses could be nociceptive reflexes, that it cannot be ruled out that they occur without the involvement of sufficiently complex central processing thought to be necessary for the animals to feel pain, and that it has not yet been conclusively demonstrated that decapods possess sensory structures of sufficient complexity to feel pain.

On the other hand, Langworthy et al. [49] reported that some decapod crustaceans have cerebral ganglia as large and well-articulated as the brains of some fish. However, this does not seem to be the case with penaeid crustaceans. Crump et al. [44] established eight criteria, four neural and four cognitive-behavioural, focusing on pain to assess the sentience of crustaceans. They found evidence that true crustaceans (Infraorder Brachyura) met five criteria, leading the authors to classify the result as solid evidence for sentience. The anomural crustaceans (suborder Anomura) and lobsters (suborder Astacidea) would meet three criteria, leading the authors to classify them as substantial evidence for sentience.

In contrast, for other suborders, such as farmed shrimps (suborder Penaeidea), the evidence for sentience is less obvious, although the authors point to the glaring gaps in knowledge that still exist for these organisms. The lack of studies specifically aimed at assessing the sentience of decapod crustaceans should, therefore, not be confused with the absence of sentience in these organisms [50]. As Birch [51] suggested, “Where there are threats of serious, negative animal welfare outcomes, lack of full scientific certainty as to the sentience of the animals in question shall not be used as a reason for postponing cost-effective measures to prevent those outcomes”.

### 1.3. Shrimp Welfare and Legislation

Even if the question of the sentience of the Penaeidea has not yet been settled among researchers, it is a fact that it is no longer possible to treat shrimps (as well as any other animal) as simple “production machines”—they are not. Whether due to scientific findings or ethical reasons, several countries have already started to enact regulations for the species-appropriate and humane slaughter of crustaceans. In Europe, although the European Union only sets rules for the slaughter of vertebrates (and excludes reptiles and amphibians) [51], the EFSA (European Food Safety Authority) concluded that decapod crustaceans could feel pain and suffering [52]. Countries such as Norway and Switzerland already include crustaceans in animal welfare legislation [53]. France enacted a “service directive” more than 10 years ago to ensure the proper handling of crustaceans before sale and consumption [54]. The Swiss government has established as a rule that only electrical stunning or “mechanical destruction” of the brain are recognised stunning methods, including for crustaceans, and requires lobsters and crayfish to be stunned before slaughter [55,56,57]. The UK government amended its Animal Welfare (Sentience) Act 2022 to include decapod crustaceans [58], which are also protected by animal welfare legislation in New Zealand [53] and Australia [58], a country that adopted the Australian Animal Welfare Strategy (AAWS) in 2004, but where the protection of crustaceans is limited to a few states.

In Asia, where some of the world’s largest shrimp producers are located, codes of conduct already include animal welfare as one of the pillars, along with food safety, animal health, and environmental integrity [33,59,60]. There is, thus, a clear tendency for the world’s largest shrimp import markets, particularly the North American and European markets, to require their suppliers to comply with norms and animal welfare standards during the production process of these organisms [20,61]. 

Thus far, the most important economic power in the world, the United States, has taken an opposite position to that of the European Union in conflicts between trade and animal welfare, prioritising economic concerns [62]. Nevertheless, due to scientific advances in recognising the sentience of decapod crustaceans or consumer demand—probably both—it will become increasingly challenging to produce shrimp without regard to animal welfare. This is due to consumer pressure, which affects all actors in the value chain, as well as issues such as sustainability certification, organic farming, and animal welfare, which have already been included in the social responsibility programmes of large companies [63,64].

The challenge, then, is to define indicators that are more measurable and less subjective that encompass different aquaculture production systems [3,65] and that focus on the welfare of farmed animals and not just the quality of the product because although these two factors are linked, they are not the same thing [4]. However, since it is impossible to ask a shrimp directly how it is doing, it is assumed that the animal’s welfare is directly related to satisfying its needs [66]. That is how the animal relates biologically, behaviourally, and emotionally to its environment [67]. Albalat et al. [21] suggested using the five domains of animal welfare (i.e., nutrition, physical environment, health, behaviour, and psychological needs) as indicators of penaeid shrimp welfare, which in turn correspond to the five freedoms established by the Farm Animal Welfare Council [68]. In this way, any measurements or observations made in a shrimp laboratory or on a shrimp farm that provide information about the extent to which the needs of the shrimp are being met can be considered potential indicators of their welfare. This paper aims to propose protocols consisting of the indicators, respective reference values and scores for assessing the welfare of *P. vannamei* in the phases of reproduction, larval rearing, transport, and growth in earthen ponds, and to discuss, based on a literature review, the processes and perspectives related to the development and application of shrimp welfare protocols. 

## 2. Materials and Methods

### 2.1. The Welfare Domains and Indicators

The indicators selected to assess the welfare of *P. vannamei* at the different stages of the production process (reproduction, larval rearing, transport, and grow-out) in earthen ponds (Figure 1) were established following the same logic already used for farmed fish species such as Atlantic salmon [69], Nile tilapia [70], and grass carp [32]. These indicators were grouped according to four of the five domains: (1) environmental, (2) sanitary, (3) nutritional, and (4) behavioural. The indicators related to psychological freedom were not considered a separate category, as the other proposed indicators assessed this freedom indirectly. 

### 2.2. Identification and Selection of Documents for Bibliographic Review and Reference Levels

The environmental, health, nutritional, and behavioural domains associated with *P. vannamei* and the indicators and their respective reference values during the reproductive, larval rearing, transport, and grow-out phases were identified based on a literature search using Google Scholar as the research platform.

Books, technical and scientific articles, case studies, manuals, and handouts developed by international institutions, theses, and dissertations were sought. The search period covered 1976 to 2023. To determine the indicators and their respective reference values, texts and documents were evaluated with the following combination of previously defined terms in their title, abstract or keywords: vannamei AND shrimp AND “pond” -biofloc AND; the specific indicators are shown in Table 1.

A database of preselected texts and documents was then set up. Subsequently, this material was analysed to evaluate the results and the methodology used by the authors to obtain them. The results were selected in situations likely to reflect biological and/or operational conditions in shrimp production.

Based on the information available in the literature, three scores were assigned (1, 2, and 3). Score 1 can be interpreted as covering the ideal variation limits for the target species. Score 2 refers to variations within the limits that animals usually tolerate. Even if such variations cause or adversely affect the animals, these are sublethal. An exception to this criterion is the indicator of mortality, which is evaluated at an intensity that can be lethal as long as it is at a lower level than in nature and, at the same time, at an intermediate level when the rearing environment is taken into account. Score 3 refers to reference levels that affect the animals’ physiological, health, and behavioural status to an unacceptable degree, so their welfare and survival are at risk.

### 2.3. Field Evaluation of the Preselected Indicators

The applicability and operational feasibility, as well as the measurement technique of each of the preselected indicators, were assessed in the field. To this end, technical visits were conducted to monitor procedures and routine management in laboratories and farms producing marine shrimp in northeastern Brazil. More specifically, we visited one farm, and one larval rearing laboratory in the state of Rio Grande do Norte (municipality of Canguaretama), three farms (in the cities of Aracati and Jaguaruana), and two larval rearing laboratories (in Aracati and Parajuru), in the state of Ceará. The companies were selected after prior contacts and expressions of interest from people responsible for improving the welfare of farmed shrimp.

The indicators that could not be collected on-site for technical or operational reasons were discarded. After this final check, the protocols were restructured in the format in which they are presented in Tables 2–14.

## 3. Results

### 3.1. Reproduction Stage

The reproductive stage ranges from the selection of animals for the formation of breeding banks to their care (in tanks or earthen ponds), to mating and spawning. According to Ceballos-Vázques et al. [71] the optimal reproductive potential of females of *P. vannamei*, based on the frequency of spawning and the quality of larvae, is reached at 12 months. As these authors point out, body weight appears to be of secondary importance at this age, although optimal breeding conditions are required for adequate growth and nutrition. This exact age is also recommended for males since the sperm quality of the animals is high from this period [72].

Eleven environmental indicators were selected to assess the welfare of shrimp-farmed breeders. In addition to the parameters commonly used to determine water quality in a farm (temperature, pH, alkalinity, ammonia, nitrite, and salinity), photoperiod (when this variable is controlled), absence of predators (terrestrial or aquatic), and stocking density were considered (Table 2). The controlled presence of terrestrial predators means that the predators are physically present in the environment but do not have direct access to the shrimp. This is the case, for example, when protective fencing prevents birds from accessing ponds. In these cases, however, even indirect contact with the predator could be detrimental to the welfare of the shrimp, e.g., as a carrier of infectious diseases. For the aquatic predators indicator, we considered controlled presence of inter-specific inhabitants in case that the producer obtained knowledge of the existence of other species in the pond, for instance, in case of polyculture.

Health indicators of *P. vannamei* breeding animals can be primarily measured by direct visual observation of the anatomical features of the animals. Luminescence, on the other hand, when observed in animals in dark environments, either in rearing facilities or in the laboratory, is indicative of the presence of bacteria of the genus Vibrio. Sexual maturity refers to the characteristics of animals that have already been selected for reproduction and spawning in the laboratory. Invasive procedures (especially removal of the eyestalk in females) are considered the most critical point in shrimp welfare at this stage of the production process and should be avoided. Mortality rates must be assessed cumulatively (from the beginning of this stage to the time of analysis or at the end of the reproductive process). Genetic selection means the application or non-application of properly standardised protocols used by the laboratory in the selection and husbandry of farmed animals (Table 3).

Nutritional indicators of *P. vannamei* breeders include, in addition to a direct indicator (the filling of the digestive tract with food), some essential aspects of the feeding routine, such as the composition and type of feed offered, the proportion of crude protein in the breeders’ artificial diet, the amount fed (as a percentage of shrimp biomass) and the feeding frequency (Table 4).

Behavioural indicators refer to shrimp swimming, feeding behaviour during management, and outcomes related to the use of stunning methods when invasive procedures are performed on the animals (Table 5).

### 3.2. Larvae Rearing Phase

According to Kitani [149], the eggs of *P. vannamei* are 0.26–0.29 mm in diameter. About 13 h after spawning, they hatch into a naupliar larva that feeds on its yolk and passes through six sub-stages (N_1_ to N_6_) before growing to about 0.46 mm in length, transforming into a zoea. At this stage, the larva feeds on phytoplankton and passes through three sub-stages (Z_1_ to Z_3_). When it reaches a size of about 2.1 mm in length, it transforms into a mysis larva. When shrimp in captivity reach this larval stage, they prefer to feed on rotifers, Artemia, or artificial food. The mysis measures up to 3.80 mm after passing through three sub-stages (M1 to M3) and swims upside down or holds its body in a vertical position with its head down. In the next stage, postlarva (PL_1_), the shrimp, about 4.2 mm long, holds its body horizontally and becomes a benthic organism. This stage continues until the animal completes its morphological development and becomes a juvenile. In a laboratory, larval rearing extends from nauplii to postlarvae when the animals are ready for marketing. 

The environmental Indicators adopted for the larval rearing stage here are essentially those already used for livestock rearing, except for the presence of predators, which are unlikely to be present in the ponds used for larval rearing. The reference values are adjusted accordingly for this life stage of the shrimp (Table 6).

The health indicators for the larvae and postlarvae of *P. vannamei* are pretty specific and concern issues related to the laboratory itself (whether or not health certificates are issued for the animals sold), the conditions in the larval rearing tanks (presence of bioluminescence in the larvae or even in the water); questions directly related to the health status of the larvae at the time of assessment (uniformity of larval stages present in the tank, presence of poorly formed larvae, presence of epibionts, muscle necrosis or melanisation of the exoskeleton, presence of lipid droplets and staining of the hepatopancreas of the postlarvae); and finally, the observed cumulative mortality rate concerning the batch analysed (Table 7).

Nutritional indicators of larval and postlarval welfare are analysed directly about shrimp feed intake (analysis of gastrointestinal tract filling) and also to diet (size of feed at each larval stage, composition of feed, and percentage of crude protein in artificial feed), as well as to the diet chosen and handling (analysis of gastrointestinal tract filling, frequency of artificial feed intake, and composition of feed), as shown in Table 8.

The behavioural indicators of larvae and postlarvae were limited to the positive phototaxic behaviour of nauplii and zoeae and the swimming activity of the animals (Table 9). 

### 3.3. Postlarvae Transport

Although it is known that in some cases, transport of nauplii takes place (sold between laboratories doing reproduction and larval culture and others doing larval culture only), only transport of postlarvae (which takes place between laboratories and farms) was considered here.

We have proposed 12 indicators for assessing the welfare of *P. vannamei* postlarvae, to be evaluated when the postlarvae arrive at the farm, including six environmental, two health, two nutritional, and two behavioural indicators (Table 10).

### 3.4. Grow-Out Stage

The growth phase begins with the transfer of postlarvae to the nurseries (in the case of biphasic rearing) or the grow-out facilities (in the case of monophasic rearing) and ends with the selection of animals for breeding banks or slaughter for marketing and consumption. The indicators proposed here and their respective reference values have been established based on rearing in earthen ponds and do not necessarily apply to other rearing systems such as bioflocs or raceways.

The environmental indicators are practically the same as those already described for the breeders, except for the photoperiod, because since the cultures are carried out in earthen ponds, the photoperiod is always the natural one. However, water transparency was included here, an indirect indicator of the number of planktonic organisms in the water (Table 11). It can be observed that the reference values for parameters such as temperature, pH, dissolved oxygen, salinity, and stocking density also differ between juveniles and breeders.

The welfare degree of the young can be assessed by anatomical indicators that can be analysed without invasive methods (in Table 12, it is suggested that producers/caretakers can identify such parameters during the biometric survey or harvest). Mortality rates can be assessed through daily monitoring, removal, and counting of dead shrimp from the pond, assessment of feed intake, and accurate quantitative surveys after harvest. 

The nutritional indicators used to assess the welfare of the juvenile are essentially indirect, except for the visual assessment of the filling of the digestive tract (Table 13). The aim is to ensure adequate feeding conditions and, thus, good nutrition for the animals. Therefore, the reference values and corresponding scores for some of the indicators (size of feed, amount of initial feed, frequency of feeding in the ponds, percentage of crude protein in the feed, and apparent feed conversion ratio) were established based on four size classes in the production process: for shrimp below 0.9 g, for juveniles from 1 to 3.9 g, and in two size classes where they can already be marketed for consumption, from 4 to 8.9 g and from 9 to 15 g. The other three indicators are independent of the size of the animals. Two scenarios were considered for the distribution of the feed: the distribution of the feed over the pond surface and the use of feeding trays. 

The behavioural indicators (Table 14) are specific to the different stages of the production process in a shrimp farm. During routine management, it is often possible to observe the swimming behaviour of shrimp, especially at the surface and edges of the ponds or near the water inlet and outlet structures. During harvesting, it is possible to assess the behaviour of the animals depending on the harvesting method used (fixed nets, at the monk’s exit, or trawls). The more restless the animals become, the more they jump (escape behaviour) and become stressed. The clinical reflexes detected during stunning are technical analyses that must be performed in detail because they are essential to assess the effectiveness of the slaughter, which is one of the most critical points for the welfare of the animals in the final grow-out stage. 

## 4. Discussion

### 4.1. Indicators for Shrimp Welfare

Sustainability of production, the environment, and animal welfare are critical issues in global food production and part of meaningful international sustainability discussions and policies, such as the United Nations Sustainable Development Goals [222]. In this context, the welfare of farmed shrimp cannot be neglected, considering how representative they are of aquaculture and how many billions of animals are produced and marketed worldwide each year [22]. 

The (few) studies on this subject still focus mainly on the reproduction and rearing/growth stages. As far as breeders are concerned, the principal reported risks are related to the removal of the eyestalk of females [21,223,224,225,226], which has several undesirable consequences, including physiological and biochemical imbalance, reproductive exhaustion, physical trauma, stress, increased energy requirements, weight loss, and increased animal mortality [21]. The main risks to shrimp welfare during the grow-out phase are related to the occurrence of diseases, especially those of viral and bacterial origin [21,227] and to the processes that eventually lead to the outbreak of these diseases, such as the problematic environmental factors related to loss of water quality or other stressful conditions to which shrimp are exposed during rearing [228,229]. Another risk to shrimp welfare that has merited the attention of researchers relates to inadequate slaughter conditions, mainly due to the absence or inadequacy of prior stunning, since immersion of animals in ice, a widely used method in shrimp slaughter worldwide, is not a stunning procedure, unlike electric shock [56]. In most publications, however, the term “shrimp welfare” now appears only sporadically and not infrequently in overly general contexts, generally referring to the most diverse aspects of the production process. 

However, for shrimp welfare to be put into practise by aquaculture producers and entrepreneurs or even included in international certification standards for companies and product marketing, concepts, and indicators that can effectively measure welfare need to be defined. Indicators of animal welfare can describe the extent to which nutritional, environmental, health, behavioural, and psychological needs are met or not met, and thus, provide information on the state of the quality of life of farm animals [227,230]. Preferably, these indicators should be relevant, valid, standardised, easy to use, reliable, comparable, and appropriate for specific systems or routines, and it is also desirable that they are simple and inexpensive [231].

We have attempted to incorporate such requirements into the indicators proposed here, following a method already proposed by our group in defining indicators of grass carp welfare [32]. In both cases, we used protocols that were always scored with a maximum of three scores (1, 2, 3) for each indicator, and we tried to define the most significant possible number of non-invasive indicators of welfare. The scores range from a positive state (score 1) to a negative one (score 3). This makes the assessment less subjective and variable compared to welfare protocols that use a larger number of scores. We also advocate for indicators of *P. vannamei* welfare that can be effectively and accurately measured and scored in the field. Indeed, some of these indicators are already integrated into routine data collection and monitoring of the production process by laboratories and shrimp farms, mainly those historically applicable through good aquaculture practises (GAP) (see examples in [232,233,234,235,236]). 

We have tested and confirmed the practical feasibility of the proposed indicators in Brazilian shrimp farms and commercial laboratories and concluded that the protocols presented here do not affect the operations of these enterprises, although they may require operational adjustments. This does not mean animal welfare protocols must be mandated in aquaculture. However, if they are difficult to apply or require high investment, they are unlikely to be adopted voluntarily by the production sector, which, as Barreto et al. [237] warned, would limit their relevance to a purely experimental context, which is not our aim. In this way, we would like to reinforce the concept that it is not incompatible in principle to adopt farming practises that promote the welfare of farmed shrimp while bringing technical, economic, and commercial benefits to aquaculture producers and entrepreneurs. 

On the other hand, we do not include physiological biomarkers that require invasive interventions among our indicators. Since there are no uniform definitions for the terms ‘invasive’ and ‘non-invasive’ when applied to animal-assisted methods [234], we assume here that “invasive” involves those procedures in which the integument of an animal is injured [37,238]. That is, we do not propose here methods in which haemolymph [239] is collected, as in the analysis of haematological [240,241], physiological [241,242,243], or biochemical [196] biomarkers of shrimp. However, this does not mean that the animals cannot be manipulated at some point to obtain relevant information. Direct observations and measurements are always essential and provide information about the welfare of the animals, as they are not themselves the cause of excessive impairment of the welfare of these animals. In the case of shrimp, invasive interventions are likely to affect their welfare seriously, so we tried to minimise them in our protocols. We believe there is always a “maximum acceptable price” to be paid for scientific knowledge. As Valente [239] questioned, is it ethically and morally correct to subject potentially sentient animals to painful or distressing tests to obtain information about their welfare? In our opinion, no. 

### 4.2. The Establishment of Farmed Shrimp Welfare Protocols

There have been several challenges in establishing indicators that reflect the welfare of *P. vannamei.* The first is related to the considerable gap that exists due to the lack of information on many aspects of the biology of this species, as pointed out by Saraiva et al. [223]. Such a gap does not mean that assumptions about animal welfare are invalid, but without it, the debate about stress or shrimp welfare runs the risk of not getting beyond the emotional or subjective level [244]. In other words, robust animal welfare protocols cannot be established without consistent scientific data. We have identified and used the most up-to-date and reliable information available in the technical and scientific literature to meet this challenge. However, we hope, recommend, and wish that the indicators proposed here, as well as the respective reference values for each of the three associated scores, will be regularly analysed and revised in the light of the knowledge that can be gained on the individual topics. 

The second challenge is related to the life stages of animals. We are trying to establish protocols that apply to all stages of the production cycle of farmed *P. vannamei* and, thus, to all stages of its life cycle, including the larval stages. However, if the understanding of the sentience of juvenile and adult shrimps is still full of doubts and gaps, what about their sentience during the larval stages? This question is not as evident as it may seem. It is already known that the shrimp’s nervous system begins to structure itself immediately after hatching. Already at the beginning of the naupliar stage (N_1_), the neuropil is formed; the anterior part of the nervous system of *P. vannamei* begins to take structural shape in N_3_ when the ganglia in the anterior part of the head coalesce; and the complete structure of the central nervous system, including proto-, deutero-, and tritocerebrum, can be seen in N_6_ [245]. Without going into the core of the debate as to whether the larval stages of *P. vannamei* are sentient or not—which is not our aim here—we have, therefore, decided to follow the precautionary principle proposed by Birc [246]: “Where there is a threat of serious negative consequences for animal welfare, the absence of complete scientific certainty about the sentience of the animals concerned should not be used as a reason for postponing cost-effective measures to avoid these consequences”. Furthermore, we believe that if we can propose methods and breeding procedures that better meet the biological needs of shrimp even at this life stage, and if we can help to ensure that producers benefit after the larval stage by including them in the GAP, why not? 

The third challenge is to recognize that the degree of tolerance or the desired level of a particular factor/indicator depends not only on the animal’s life stage but also on the production system used and the environmental conditions in which they are kept [21]. In this article, due to space constraints, it is neither possible nor necessary to discuss individually all the indicators proposed here for the welfare of *P. vannamei* at the different breeding stages, as both the indicators and the reference values are intended to be self-explanatory. However, it should be emphasised that the welfare indicators almost always interact directly and indirectly. In the following, we present two classic cases that illustrate this: the indicators “salinity” and “stocking density”.

*Penaeus vannamei* is known as one of the euryhaline shrimps [247]. Their reproduction takes place in oceanic waters (with high salinity), while postlarvae are settled in coastal areas (estuaries and lagoons with variable salinity) [248], where they arrive after developing their physiological abilities to overcome the fluctuating nature of these environments [249]. In these coastal regions, animals grow throughout the juvenile stage before returning to marine areas, where they reach sexual maturity and reproduce [248]. It is known that the mechanisms of euryhalinity and osmoregulation change during the ontogenetic development of *P. vannamei*. The hyper-hypo osmoregulatory pattern exhibited by juveniles and adults appears to be established at the beginning of the first postlarval stage (PL_1_), with greater tolerance to salinity fluctuations observed in PL_2_, PL_4_, and PL_22_, suggesting that *P. vannamei* exhibits a progressive increase in the efficiency of the osmoregulatory mechanism after the last larval stage [247]. However, it is known that salinity is not only an indicator that directly affects growth, survival [250], and, thus, the welfare of the species, but that in a husbandry system, it also directly affects changes in other parameters that determine water quality and thus shrimp welfare, such as solubility of oxygen [251]. At the same time, salinity can also influence the oxygen consumption of the shrimp themselves [252] and even their tolerance to low dissolved oxygen concentrations [253]. 

The stocking density in the rearing phase is admittedly a critical decision in farmed shrimp production [204,205]. The density, and thus, the sustainable biomass to be achieved, is closely related to the carrying capacity of the chosen system and varies considerably between different farming systems [254]. In extreme cases, shrimp may arrive at a farm and initially be kept in nursery tanks with densities exceeding 1500 postlarvae/m^2^ [255] and then exceed 400 shrimp/m^2^ during the final stages if the chosen husbandry system is Biofloc [256], or exceed 150 shrimp/m^2^ in earthen ponds [257]. From an animal welfare perspective, it is recommended that the maximum stocking density in ponds during the rearing phase does not exceed 50 shrimp/m^2^ [205]. Higher stocking densities tend to increase the stress to which shrimp are subjected, which from a technical point of view can compromise the results to be achieved by the operator (e.g., by reducing and compromising growth and survival rates, feed consumption, and final biomass achieved) and from the animals’ point of view can severely damage their immune system [258] affecting their welfare [21]. The expression of a particular disease—another critical issue for the welfare of farmed shrimp—is almost always the result of a complex interplay of factors related not only to stocking density but also to intrinsic aspects of the cultivated species, the pathogen in particular, and the environment [259]. This requires that disease control programmes take a multifactorial approach and that any innovation introduced should be seen as just one more element in this complex process [19].

The ecological and biological variables are so dynamic and interactive that we believe it is impossible to think of standardised management techniques or control environmental variables in ponds and hatcheries for shrimp farming. On the other hand, it is perfectly possible, for example, to establish acceptable ranges of variation in environmental parameters; to ensure the supply of natural food to the animals; to offer pre- and probiotics as part of the shrimp diet; to stimulate the growth of beneficial microbial communities; to carry out regular tests on the health status of the farmed animals; to promote their stunning before slaughter. Our protocols aim to help measure the impact of all these practises on the welfare of farmed *P. vannamei*.

## 5. Conclusions

The welfare of farmed shrimp has only recently attracted the interest of researchers, producers, policy makers, and other stakeholders in this valuable production chain [260]. As we noted in our literature review, this paper is the first attempt to propose indicators that encompass all stages of the production process of a shrimp species and, in this case, an essential shrimp species for global aquaculture. However, it is foreseeable that technologies for monitoring the welfare of farmed shrimp and those specifically targeting GAP will increasingly converge from now on. This development will typically take place through so-called “precision aquaculture” [256], which will include technologies such as the use of biosensors, data loggers, and early warning systems [21]; computer vision for animal monitoring, sensor networks (wireless and long range), robotics, and decision support tools, such as algorithms, the Internet of Things, and decision support systems [261,262,263]; as well as the use of high-quality and sensitive cameras for automatic detection and analysis of shrimp behaviour even in turbid waters [145,264]. Such technologies will, in turn, provide shrimp farmers with important information to manage feed supply with minimum waste and maximum feed efficiency; assess organic waste accumulation in ponds to optimise the water quality available to shrimp; improve management decisions related to animal health; and increasingly use animal behavioural signals as indicators of their welfare and the efficiency of management practices applied.

Likely, these non-invasive methods of measuring shrimp welfare will soon become routine in farms and laboratories, and it will become increasingly challenging to produce shrimp without taking into account the welfare of the organisms that are farmed, slaughtered, and offered to consumers, whether due to scientific advances in decapod sensitivity, consumer market demand, or changes in international regulations governing shrimp production and marketing—or all of these factors combined. The best evidence that this is entirely plausible is that remote water quality monitoring technologies are already a reality in several places worldwide [265,266,267,268]. Further evidence is that large companies are already incorporating issues such as sustainability certification, organic farming, and animal welfare into their social responsibility programmes [63,64]. Although the possible technological revolution has the potential to facilitate the assessment of welfare indicators, it will not render the use of these indicators superfluous. On the contrary, as is already the case with remote water quality monitoring, there is a tendency to integrate it with this and other technological tools. However, until this happens, any shrimp farmer can already measure the welfare of *P. vannamei* during the different stages and breeding processes.

## Figures and Tables

**Figure 1 animals-13-00807-f001:**
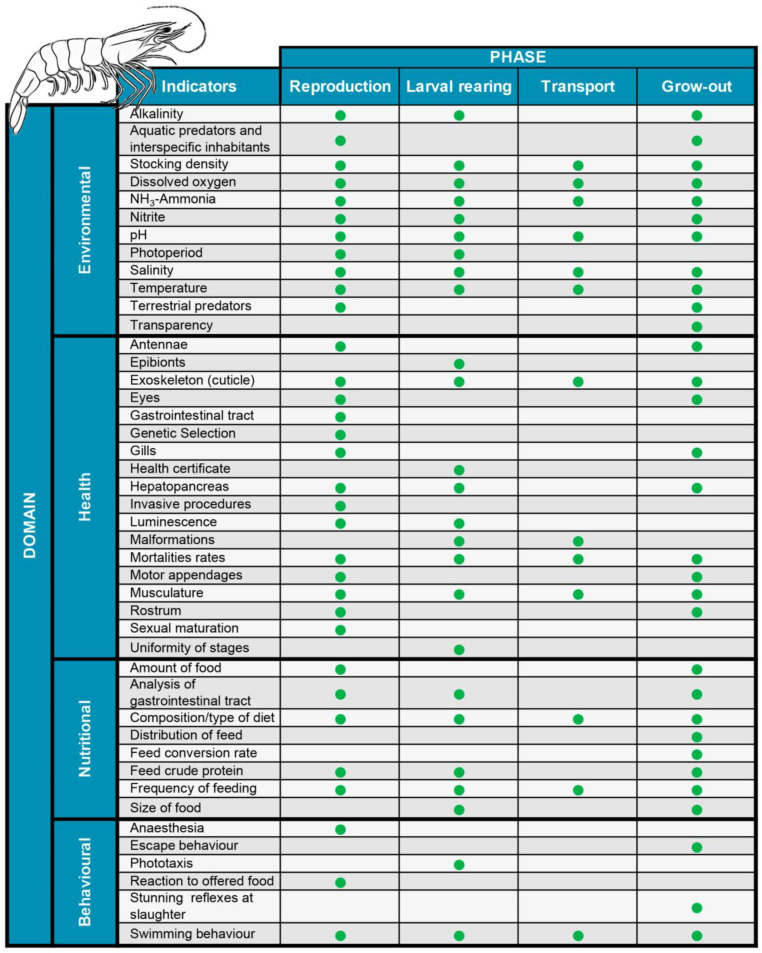
Domains and welfare indicators during the production process of white-leg shrimp, *Penaeus vannamei*.

**Table 1 animals-13-00807-t001:** Specific indicators used to obtain bibliographic data to define the reference values for the four domains of marine shrimp welfare studied.

Domain	Specific Indicators
Environmental	water quality; temperature; pH; transparency; dissolved oxygen; ammonia; nitrite; salinity; alkalinity; predators; competitors; photoperiod; stocking density
Health	health; diseases; eyes; exoskeleton; gills; hepatopancreas; antennae; rostrum; mortality
Nutritional	protein; feeding frequency; feed amount; food distribution
Behavioural	behaviour; respiratory frequency; swimming; feed intake; anaesthesia; handling; slaughter

**Table 2 animals-13-00807-t002:** Environmental indicators for *Penaeus vannamei* breeders.

Indicators	Score	Reference Values	References
Temperature (°C)	1	24.5–32.5	[73,74,75,76,77,78,79,80]
	2	15.6–24.4 or 32.6–35.4	
	3	≤15.5 or ≥35.5	
pH	1	7.5–8.5	[75,76,77,79,81,82,83,84]
	2	5.0–7.4 or 8.6–9.0	
	3	≤4.9 or ≥9.1	
Photoperiod (Light: Dark)	1	Natural or 12L:12D–14L:10D	[75,76,79,80,85,86]
	2	15L:9D–16L:8D	
	3	17L:7D or clearer; 11L:13D or darker	
Alkalinity (mg/L CaCO_3_)	1	100–140	[81,83,87,88,89]
	2	51–99 or 141–199	
	3	≤50 or ≥200	
Dissolved oxygen (% saturation)	1	≥62	[80,81,83,87,88,90,91]
	2	46–61	
	3	≤45	
Non-ionised ammonia (mg/L NH_3_)	1	0.00–0.10	[81,87,88,89,92,93]
	2	0.11–0.30	
	3	≥0.31	
Nitrite (mg/L NO_2_^−^)	1	0–0.6	[81,83,87,88,89,92,94,95]
	2	0.7–1.6	
	3	≥1.7	
Salinity (psu)	1	28–35	[75,76,78,79,84,96,97]
	2	10–27 or 36–40	
	3	≤9 or ≥41	
Stocking density (g/m^2^)	1	≤150	[76,79,98,99,100]
	2	151–299	
	3	≥300	
Terrestrial predators *	1	Absence	[101,102]
	2	Controlled presence	
	3	Uncontrolled presence	
Aquatic predators and interspecific inhabitants **	1	Absence	[102,103,104]
	2	Controlled presence	
	3	Uncontrolled presence	

* Birds, mammals, and reptiles. ** Fish, other crustaceans (crabs and shrimps), molluscs (snails), amphibians (frogs), and reptiles (water snakes).

**Table 3 animals-13-00807-t003:** Health indicators for *Penaeus vannamei* breeders.

Indicators	Score	Description or Reference Values	References
Antennae	1	Healthy appearance, no changes	[105,106,107]
	2	Focal lesion, shortening or darkening	
	3	Absence, blueness, wrinkling, multifocal dark spots	
Rostrum	1	Healthy appearance, no changes	[107,108,109]
	2	Mild injury, erosion or necrosis	
	3	Severe injury, erosion or necrosis, deformity, bending to one side, upwards or downwards	
Eyes	1	Healthy appearance, no changes	[108,110,111]
	2	Unilateral lightening, injury, softening or swelling	
	3	Bilateral lightening, injury, softening or swelling, absence of one or both organs	
Gills	1	Healthy appearance, no changes	[112,113,114]
	2	Focal lesion or darkening	
	3	Pale, yellowish, general redness, darkening, whitish spots, or erosion	
Hepatopancreas	1	Healthy appearance, no changes	[107,112,115,116]
	2	Discrete volume reduction	
	3	Atrophy, stiffness, flaccidity, colour change (abnormal pallor or darkening, appearance of streaks of different shades), volume change, tearing, presence of worm-like structures, presence of fluid (oedema)	
Motor appendages (pereiopods/uropods/pleopods)	1	Healthy appearance, no changes	[112,115,117]
	2	Focal absence or erosion	
	3	Severe or complete absence, colour change, necrosis (lightening), dark spots, rough and darkened edges	
Exoskeleton (cuticle)	1	Healthy appearance, no changes	[88,112,115]
	2	Slight lesion or focal darkening, presence of debris	
	3	Tissue loss, necrosis, focal or generalised colour change, deformity, calcified round white patches, crusting	
Musculature	1	Healthy appearance, no changes	[105,108,117]
	2	Focal necrosis (lightening)	
	3	Generalised necrosis, colour change (yellowing, redness, opacity, milky appearance), atrophy, swelling, stiffness (constant bending), zigzag shape	
Gastrointestinal tract	1	Healthy appearance, no changes	[88,108,118]
	2	Presence of sand particles or debris	
	3	Atrophy, empty, pale, whitish bowel, whitish stools, faecal streaks	
Luminescence	1	No luminescence is observed in breeders in complete darkness	[79]
	2	Not applicable	
	3	Luminescence is observed in breeders in absolute darkness	
Sexual maturation	1	Females: Enlarged ovaries, distinct olive-green colour (stage IV). Males: Shiny sperm ampullae	[75,81]
	2	Not applicable	
	3	Females: Small, translucent ovaries (stage I); opaque and yellowish (stage II); bright yellow (stage III); spawned (stage V). Males: opaque or brownish (necrotic or melanised) sperm ampullae	
Invasive procedures (ablation of the eyestalk, extrusion of the spermatophore) *	1	No invasive intervention	[75,119,120,121]
	2	Invasive procedure with effective anaesthesia and postoperative analgesia	
	3	Invasive procedure without effective anaesthesia and postoperative analgesia	
Mortality during breeding and spawning (%) **	1	≤10	[70,122,123]
	2	11–25	
	3	≥26	
Genetic selection	1	Genetic selection and or improvement programme with inbreeding control	[124]
	2	Genetic selection and or improvement programme without inbreeding control	
	3	No genetic improvement programme	

* There is not yet sufficient scientific evidence that the signs known as surgical staging signs cause adequate analgesia and unconsciousness in shrimp. ** More research is needed in order to understand how to decrease larval mortality in captive shrimp.

**Table 4 animals-13-00807-t004:** Nutritional indicators for *Penaeus vannamei* breeders.

Indicators	Score	Reference Values	References
Filling of the digestive tract *	1	Full	[125]
	2	Medium	
	3	Empty	
Frequency of feeding (times/day)	1	≥3	[87,126,127,128,129,130,131,132]
	2	2	
	3	≤1	
Composition/type of diet	1	Special artificial feed for breeders + fresh feed	[133,134,135]
	2	Special artificial feed for breeders only	
	3	Fresh feed only	
Amount of food (% of biomass **)	1	≥4.0	[75,136,137,138,139,140]
	2	2.4–3.9	
	3	≤2.3	
Crude protein in the breeders’ artificial diet (%)	1	≥35	[128,137,140,141,142,143]
	2	29–34	
	3	≤28	

* Observation after a minimum of 30 min and a maximum of 2 h after offering the food. ** Fresh + artificial feed.

**Table 5 animals-13-00807-t005:** Behavioural indicators for *Penaeus vannamei* breeders.

Management	Indicators	Score	Description	References
Routine management	Swimming behaviour	1	Animals at the bottom that stop, walk or feed, or normally swim in the water column	[91,144,145]
		2	Few animals with irregular swimming or with partial loss of balance at the tank bottom	
		3	Many animals with irregular swimming (off balance or in a spiral), crowding of shrimp in certain parts of the tank	
Feeding behaviour	Reaction to offered food	1	Most animals react quickly by moving to the newly offered food	Authors’ suggestion
		2	Only some of the animals react to the newly offered food	
		3	Most animals do not react to the newly offered food	
Invasive methods	Anaesthesia—surgical phase * (loss of balance and response to external stimuli)	1	Induction and recovery in 3–5 min	[146,147,148]
		2	Induction and or recovery in 6–9 min	
		3	Absence of anaesthetic procedures Induction and or recovery ≤ 2 min or ≥10 min; death	

* There is not yet sufficient scientific evidence that the signs known as surgical staging signs cause adequate analgesia and unconsciousness in shrimp.

**Table 6 animals-13-00807-t006:** Environmental indicators for larvae and postlarvae of *Penaeus vannamei*.

Indicators	Score	Reference Values	References
Temperature (°C)	1	26.5–32.4	[79,99,149,150,151,152,153]
	2	19.5–26.4 or 32.5–35.4	
	3	≤19.4 or ≥35.5	
pH	1	7.5–8.5	[79,99,149,151]
	2	5.0–7.4 or 8.6–9.0	
	3	≤4.9 or ≥9.1	
Photoperiod (Light:Dark)	1	18L:6D–24L:0D	[86,154,155]
	2	17L:7D–12L:12D	
	3	11L:13D or darker	
Alkalinity (mg/L as CaC O_3_)	1	100–140	[81,83,87,88,89]
	2	51–99 or 141–199	
	3	≤50 or ≥200	
Dissolved oxygen (% saturation)	1	≥64	[79,99]
	2	49–63	
	3	≤48	
Non-ionised ammonia (mg/L NH_3_)	1	0.00–0.01	[99,156,157,158]
	2	0.02–0.06	
	3	≥0.07	
Nitrite (mg/L NO_2_^−^)	1	0.0–0.1	[99,157,159]
	2	0.2–0.7	
	3	≥0.8	
Salinity (psu)	1	30–36	[79,99,149,150,151,152]
	2	20–29 or 37–44	
	3	≤19 or ≥45	
Stocking density (Larvae or postlarvae/L)	1	≤250 (larvae) ≤100 (postlarvae)	[79,160,161]
	2	251–300 (larvae) 101–210 (postlarvae)	
	3	≥301 (larvae) ≥211 (postlarvae)	

**Table 7 animals-13-00807-t007:** Health indicators for larvae and postlarvae of *Penaeus vannamei*.

Indicators	Scores	Reference Values or Description	References
Health certificate	1	SPF ^1^ + SPT ^2^ and or SPR ^3^	[79,88]
	2	SPF ^1^	
	3	Does not have a health certificate	
Luminescence	1	No luminescence is observed in the larvae tank under absolute darkness	[79]
	2	Not applicable	
	3	luminescence is observed in the larvae tank under absolute darkness	
Uniformity of larval stages in a tank *	1	≥75% of the sampled larvae are in the same larval stage	[79,88]
	2	50–74% of the sampled larvae are in the same larval stage	
	3	≤49% of the sampled larvae are in the same larval stage	
Malformations (%)	1	Low (≤5% of the sampled larvae have deformities)	[79,88,152]
	2	Moderate (6–10% of the sampled larvae have deformities)	
	3	Severe (≥11% of the sampled larvae have deformities)	
Staining of the hepatopancreas *	1	≥90% of larvae have darkened hepatopancreas	[79,152,162]
	2	70–89% of larvae have darkened hepatopancreas	
	3	≤69% of larvae have darkened hepatopancreas	
Condition of the hepatopancreas *	1	≥90% of sampled larvae have abundant lipid vacuoles	[79,152,162]
	2	70–89% of sampled larvae have abundant lipid vacuoles	
	3	≤69% of the sampled larvae have abundant lipid vacuoles	
Epibiont encrustation on the exoskeleton, appendages and gills *	1	≤5% of sampled larvae have fouling	[79,152]
	2	6–10% of sampled larvae have encrustations	
	3	≥11% of sampled larvae suffer from fouling	
Necrosis * (%)	1	Absence of necrosis in larvae	[79,88]
	2	≤15% of larvae show opacities of the muscles or limbs	
	3	≥16% of larvae show necrosis in muscles or limbs	
Melanisation * (%)	1	≥90% of larvae show up to 5% melanisation	[79]
	2	70–89% of larvae show up to 5% melanisation	
	3	≤69% of larvae show up to 5% melanisation	
Mortality at the end of larval rearing **	1	≤30	[79,163,164,165]
	2	31–49	
	3	≥50	

^1^ SPF: Specific pathogen-free postlarvae. Postlarvae must be negative for YHV, IHHNV, WSSV, TSV, IMNV, and NHP according to PCR analyses. ^2^ SPT: Specific pathogen tolerant postlarvae. ^3^ SPR: Specific pathogen-resistant postlarvae. * These analyses require a magnifying glass or microscope and should only be carried out if this is already part of the laboratory’s quality control. ** More research is needed in order to understand how to decrease larval mortality in captive shrimp.

**Table 8 animals-13-00807-t008:** Nutritional indicators for larvae and postlarvae of *Penaeus vannamei*.

Indicators	Score	Reference Values			References
		0.002–0.08 g	0.081–1.0 g	1.1–2.5 g	
Size of the food (µm)	1	Fed mash-700	701–1800	1801 µm –3.2 mm	[125,148,166]
	2	≥701	≤701	≤1801	
	3	Not applicable	≥1801	≥3.3	
Crude protein in the artificial feed (%)	1	≥25	≥25	≥40	[140,167,168,169,170]
	2	20–24	20–24	25–40	
	3	≤19	≤19	≤24	
Analysis of PL gastrointestinal tract (% of sampled animals with complete tract)	1	Does not apply	Does not apply	≥90	[79]
	2			70–89	
	3			≤69	
Frequency of feeding—artificial feed (times/day)	1	≥6			[87,171,172]
	2	3–5			
	3	≤2			
Composition of the diet (type of food)	1	Specific artificial food for each larval stage + live food			[173,174,175,176,177]
	2	Only specific artificial food for each larval stage			
	3	Only live food			

**Table 9 animals-13-00807-t009:** Behavioural indicators for larvae and postlarvae of *Penaeus vannamei*.

Indicators	Score	Description or Reference Values	References
Phototaxis of nauplii and zoea	1	≥95% of sampled larvae are attracted to the light from the water surface	[79,152]
	2	70–94% of sampled larvae are attracted to the light from the water surface	
	3	≤69% of sampled larvae are attracted to the light from the water surface	
Swimming behaviour of larvae and postlarvae	1	≥95% of the sampled larvae are active	[79,152,162]
	2	70–94% of the sampled larvae are active	
	3	≤69% of the sampled larvae are active	

**Table 10 animals-13-00807-t010:** Indicator for the transport of postlarvae of *Penaeus vannamei*.

Category	Indicators	Score	Description or Reference Values	References
Environmental	Temperature (°C)	1	22.5–25.4	[81,88,149,178,179]
		2	19.5–22.4 or 25.5–29.5	
		3	≤19.4 or ≥29.6	
	pH	1	6.5–8.0	[178]
		2	5.0–6.4 or 8.1–8.5	
		3	≤4.9 or ≥8.6	
	Salinity (psu)	1	Acclimatised to the salinity of the farm	[79,99,149,150,151,152]
		2	Need to acclimatise for up to 5 ups	
		3	Need to acclimatise for more than 5 ups	
	Dissolved oxygen (% saturation)	1	≥80	[81]
		2	60–79	
		3	≤59	
	Transport density (postlarvae/L) *	1	≤750	[81,179,180,181]
		2	751–1000	
		3	≥1001	
	Non-ionised ammonia (mg/L NH_3_)	1	0.00–0.01	[99,156,157,158]
		2	0.02–0.06	
		3	≥0.07	
Health	Lesions, necrosis and/or malformations (rostrum, appendages and or gills) (%)	1	≤5	[182]
		2	6–10	
		3	≥11	
	Mortality (%)	1	≤5	[178,180]
		2	6–10	
		3	≥11	
Nutritional	Feeding (time interval) **	1	≤3 h during transport	[88,149,179,183,184]
		2	≥4 h during transport	
		3	No feeding during transport	
	Type of food added	1	Artemia nauplii + industrial feed	[179,183]
		2	Only industrial feed or artemia nauplii	
		3	No additional feed during transport	
Behavioural	Swimming behaviour after transport (no anaesthesia) ***	1	≥95% of active postlarvae and normal swimming ability	[79,152,162,185,186]
		2	70–94% of active postlarvae and normal swimming ability	
		3	≤69% of active postlarvae and normal swimming ability	
	Swimming behaviour after transport (with anaesthesia)	1	Sedation (loss of balance with reaction to external stimuli)	[146,187]
		2	Anaesthesia (complete loss of equilibrium with no response to external stimuli)	
		3	No induction or recovery, death	

* Stocking density applies to PL_10_ to PL_20_ in shipping bags or transport crates. ** Offering food during transport is done when using transport boxes. When transporting plastic bags, there is no possibility of offering food after the bags have been sealed. *** Abnormal swimming includes jumping at the surface, swimming at an angle, swimming in figure eights, swimming at the surface, and no swimming.

**Table 11 animals-13-00807-t011:** Environmental indicators for juvenile and adult *Penaeus vannamei.* It is recommended that postlarvae are previously acclimatised in ponds for 10 to 15 min for every degree of temperature difference and salinity unit (81; 149) and at least 60 min per different 0.5 pH unit (81; 88).

Indicators	Score	Reference Values	References
Temperature (°C)	1	25.5–32.4	[82,88,188,189,190,191,192,193,194]
	2	14.5–25.4 or 32.5–35.4	
	3	≤14.4 or ≥35.5	
pH	1	6.5–8.5	[81,82,83,88,195,196]
	2	5.0–6.4 or 8.6–9.0	
	3	≤4.9 or ≥9.1	
Transparency (cm)	1	35–50	[81,87,88,89,90]
	2	21–34 or 51–59	
	3	≤20 or ≥60	
Alkalinity (mg/L CaCO_3_)	1	100–140	[81,83,87,88,89]
	2	51–99 or 141–199	
	3	≤50 or ≥200	
Non-ionised ammonia (mg/L de NH_3_)	1	0.00–0.10	[81,87,88,89,92,93,197]
	2	0.11–0.30	
	3	≥0.31	
Dissolved oxygen (% saturation)	1	≥65	[81,83,87,88,90,91]
	2	49–64	
	3	≤48	
Nitrite (mg/L NO_2_)	1	0.0–0.6	[81,83,87,88,89,92,94,95]
	2	0.7–1.6	
	3	≥1.7	
Salinity (psu)	1	10.0–40.9	[81,87,88,91,192,194,198]
	2	0.6–9.9 or 41.0–59.0	
	3	≤0.5 or ≥60.0	
Stocking density (shrimp/m^2^)	1	≤40	[199,200,201,202,203,204,205]
	2	41–60	
	3	≥61	
Terrestrial predators *	1	Absence	[101,102]
	2	Controlled presence	
	3	Uncontrolled presence	
Aquatic predators and interspecific inhabitants **	1	Absence	[102,103,104]
	2	Controlled presence	
	3	Uncontrolled presence	

* Birds, mammals, and reptiles. ** Fish, other crustaceans, molluscs, amphibians, and reptiles.

**Table 12 animals-13-00807-t012:** Health indicators for *Penaeus vannamei* juveniles.

Indicators	Score	Description or Reference Values	References
Antennae	1	Healthy appearance, no changes	[105,106,107]
	2	A focal lesion, shortening, or darkening	
	3	Absence, blueness, wrinkling, multifocal dark spots	
Rostrum	1	Healthy appearance, no changes	[107,108,109]
	2	Mild injury, erosion, or necrosis	
	3	Severe injury, erosion or necrosis, deformity, bending to one side, upwards or downwards	
Eyes	1	Healthy appearance, no changes	[108,110,111]
	2	Unilateral lightening, injury, softening or swelling	
	3	Bilateral lightening, injury, softening or swelling, absence of one or both organs	
Gills	1	Healthy appearance, no changes	[112,113,114]
	2	Focal lesion or darkening	
	3	Pale, yellowish, general redness or darkening, whitish spots, erosion	
Hepatopancreas	1	Healthy appearance, no changes	[107,112,115,116]
	2	Discrete volume reduction	
	3	Atrophy, stiffness, flaccidity, colour change (abnormal pallor or darkening, appearance of streaks of different shades), volume change, tearing, presence of worm-like structures, presence of fluid (oedema)	
Motor appendages (pereiopods/uropods /pleopods/telson)	1	Healthy appearance, no changes	[112,115,117]
	2	Focal absence or erosions	
	3	Severe or complete absence, colour change, necrosis (lightening), dark spots, rough and darkened edges	
Exoskeleton (cuticle)	1	Healthy appearance, no changes	[88,112,115]
	2	Slight lesion or focal darkening, presence of debris	
	3	Tissue loss, necrosis, focal or generalised colour change, deformity, calcified round white spots, encrustation	
Musculature	1	Healthy appearance, no changes	[105,108,117]
	2	Focal necrosis (lightening)	
	3	Generalised necrosis, colour change (yellowing, redness, opacity, milky appearance), atrophy, swelling, stiffness (constant bending), zigzag shape	
Mortality (%) *	1	≤10	[70,122,123]
	2	11–25	
	3	≥26	

* More research is needed in order to understand how to decrease larval mortality in captive shrimp.

**Table 13 animals-13-00807-t013:** Nutritional indicators for *Penaeus vannamei* juveniles and adults.

Indicators		Score	Weight (g)				References
			≤0.9	1.0–3.9	4.0–8.9	9.0–15.0	
Size of food (mm)		1	0.1–0.5	0.6–1.0	1.1–2.0	2.1–3.0	[149,206]
		2	≥0.6	≥1.1	≥2.1	≥3.1	
		3	<0.1	≤0.5	≤1.0	≤2.0	
Amount of initial food (% of biomass)		1	6.0–10.9	4.0–6.9	4.0–6.9	2.0–3.9	[75,87,126,206,207]
		2	4.1–5.9	2.1–3.9	2.1–3.9	1.1–1.9	
		3	≤4.0 or ≥11.0	≤2.0 or ≥7.0	≤2.0 or ≥ 7.0	≤1.0 or ≥4.0	
Frequency of feeding in the ponds (times/day)		1	≥4	≥2	≥2	≥2	[87,126,127,128,129,130,131,132,208]
		2	2–3	1	1	1	
		3	≤1	<1	<1	<1	
Feed crude protein (%)		1	≥35	≥35	≥32	≥32	[87,126,127,128,140,142,206]
		2	32–34	32–34	25–31	25–31	
		3	≤31	≤31	≤24	≤24	
Apparent feed conversion rate (FCR) *		1	Does not apply	Does not apply	≤1.5	≤1.7	[89,126,209,210,211,212,213]
		2			1.6–2.0	1.8–2.0	
		3			≥2.1	≥2.1	
Feed distribution	Distribution of feed over the surface (% of pond surface) OR	1	≥75				[87,127,207,209,214,215,216]
		2	50–74				
		3	≤49 or no control of coverage area				
	Use of trays (number of trays/ha)	1	≥20				
		2	15–19				
		3	≤14				
Digestive tract filling index **		1	Full				[217]
		2	Medium				
		3	Empty				

* Calculated at the end of culture for shrimps with an average of 15 grammes. FCR = 100 × [ln (FW − IW)/T]: FW: final weight; IW: Initial weight; T: Time interval (days). ** Observation after a minimum of 30 min and a maximum of 2 h after offering the feed.

**Table 14 animals-13-00807-t014:** Behavioural indicators for *Penaeus vannamei* juveniles and adults.

Management	Indicators	Score	Reference Values	References
Routine management	Swimming behaviour	1	No shrimp on the pond surface or irregular swimming	[91,144,145]
		2	Few animals on the pond surface or irregular swimming	
		3	Reduced, irregular or “spiral” swimming, accumulation of shrimp at the edges of the pond or near the water inlet, many animals exposing their bodies at the water surface	
Partial or complete harvesting	Escape behaviour (successive tail movements by flexion and extension of the abdomen)	1	Few jumping shrimps during harvest, with low frequency and intensity	[218]
		2	Few jumping shrimps, but with high frequency and/or intensity during harvesting	
		3	Many jumping shrimps, high frequency and/or intensity during harvesting	
Stunning at slaughter *	Clinical reflexes	1	Immediate loss of response to external stimuli; balance (with cephalothorax in horizontal and descending position); movement of pleopods and pereiopods; and movement of scaphognathites	[146,147,186,219,220,221]
		2	Progressive loss of response to external stimuli; balance (with cephalothorax in horizontal and descending position); movement of pleopods and pereiopods; and movement of scaphognathites in ≤30 s	
		3	Progressive loss of: Response to external stimuli; balance (with cephalothorax in horizontal and descending position); movement of pleopods and pereiopods; and movement of scaphognathites in >30 s	

* The use of ice or simple exposure to air is not considered a stunning method. Therefore, the treatment is given a score of 3.

## Data Availability

Not applicable.

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
