# Peer review of "Non-Invasive Methods for Assessing the Welfare of Farmed White-Leg Shrimp (Penaeus vannamei)"

_animals, 2023, doi:10.3390/ani13050807_

Round 1

Reviewer 1 Report

Despite its limitations – in part due to important gaps in knowledge, the authors make a valuable effort to bring attention to the welfare of shrimp in captivity, and provide operational tools to investigate the welfare of these animals in aquaculture scenarios. This piece of research has the potential to generate fruitful scientific and ethical debates regarding shrimp aquaculture, and stimulate follow up research projects aiming at addressing its weaknesses and existing knowledge gaps.

Comments and suggestions:

L. 26: I would use the term “animal welfare” instead of “animal experience”. The former is clearer from a scientific viewpoint. 

Ll. 49-50: It is not clear to me what do the authors mean by “one of the most agri-food systems for animal protein production in the world”.

L. 80: I suggest replacing “Those related to the sentient (sic) of these animals“ with “ Those related to the current scientific evidence for sentience in these animals”.

Ll. 84-85:  I suggest replacing “the welfare of an organism is inseparable from the absence of suffering and the creation of positive conditions for that suffering” with “the welfare of an organism is inseparable from the degree of suffering and the positive states that this individual experiences at a certain point in time”.

Ll. 85-86: I suggest deleting the sentence “However, according to most national and international policy frameworks, these experiences only apply to sentient animals, as non-sentient animals cannot feel.” The reasoning of the authors is confusing to me, since policy frameworks are not relevant in terms of defining sentience – in fact, they are actually crafted taking into consideration the scientific evidence available. Also, the sentence is not relevant to the topic discussed in this section.

Ll. 103-106: The authors mention higher vertebrates but provide examples of invertebrate groups. Please correct and clarify.

L.192: Figure 1. Have the authors considered exploring the aspect of environmental design (within the tank or pond) and potential environmental enrichment? Same comment applies to Table 1. below? Also, this is a potential good point to add to the discussion.

L. 205: Table 1. “Stocking density” instead of “storage density”.

L. 241: Table 2. Three different colours respectively indicating Scores 1, 2, and 3 would make the interpretation of the table more straightforward. Same comment applies to Tables 2-14. 

L. 247: “Stocking density” instead of “Storage density”.

L. 251: It would be clearer if the authors specify what they mean by “controlled presence”. Are predators present but effectively controlled? Are predators controlled but still killing/injuring a certain percentage of individuals whose welfare is therefore negatively impacted?

L. 273: Table 3. From an operational viewpoint, it would be very useful if the authors could provide footnotes/asterisks with more specifics about the characteristics of the lesions (e.g., gills’ focal lesion or darkening - approximate size of the lesion?; hepatopancreas’ discrete volume reduction - approximate percentage?; etc.). This comment also applies to the following tables. 

L. 273: Table 3. Invasive procedures (ablation of the eyestalk, extrusion of the spermatophore) / The invasive procedure with anaesthesia - Perhaps mention instead “effective anaesthesia and postoperative analgesia, and provide a footnote with its characteristics regarding the case of female breeders? If the asterisk note in line 291 also applies to this row, it would be important to mention it here too. 

Tables 2-14 - it would be very informative if the researchers could create an additional column with several categories indicating their opinion regarding the strength of their findings (according to the existing evidence, e.g., strong/medium/weak). That would be extremely helpful in order to understand the current knowledge gaps (and stimulate research projects to fill those gaps). The authors could further elaborate about this point (additional column) as part of Section “4.2. The establishment of farmed shrimp welfare protocols”.

L. 313: Table 7. Mortality at the end of larval rearing. Does almost 50% mortality qualify as Score 2 (described as “sublethal)? Perhaps this is not a problem for the species or the producer, but it looks like a very problematic welfare issue at the individual level (if we consider that the welfare of shrimp larvae must be taken into account, obviously – the authors make a good point about this point as part of their discussion section). 

Also, their conclusion regarding mortality percentages is supported by just one paper that is mostly focused on productivity (therefore, an additional column informing about the strength of the findings would be of great help – please see my previous comment). If additional sources are not available, it would be important to add a footnote/asterisk mentioning that more research is needed in order to understand how to decrease larval mortality in captive shrimp. This comment also applies to the mortality sections in the different tables, since mortality percentages under current productive conditions are likely different from what is possible under improved conditions in captivity (the gold standard we must work towards in terms of providing good welfare). 

L. 414: Table 14. Stunning at slaughter - it would be important to mention the duration of the stun before the animals regain consciousness (if this information is available). Typically, this is a problem with electrical methods of stunning, and for obvious welfare reasons the animals should be killed before they become conscious.

Reviewer 2 Report

This paper identifies and characterises effective, measurable, non-invasive indicators based in the behavioural, nutritional, health and environmental domains for assessing the welfare of farmed white-leg shrimp (Penaeus vannamei) during reproduction, larval rearing, transport and growth. Non-invasive is defined as methods which do not injure the integument, so excludes collection of samples such as haemolymph for haematological and biochemical analyses. These indicators aim to be simple, inexpensive, relevant, valid, standardised, easy to use and reliable in order to encourage uptake by the production sector. The paper explains why welfare in these animals has been neglected and provides convincing scientific, social and legal arguments for why it will become unavoidable. It is clear, logical and thoroughly researched.

Comments

The indicators selected to assess the welfare of P. vannamei are given three possible values for animal experience. These are described as ranging from positive (score 1) to very negative (score 3) (line 26) or a positive state (score 41) to a negative one (score 3) (line 455).

Lines 212-217 explain these scores cover the ideal indicator range for the target species (score 1), variations which are tolerable (score 2) and reference values that significantly influence the physiological, health and behavioural status of animals (score 3).

It may be helpful to include the terms wellbeing (score 1), stress (score 2) and distress (score 3) in the explanation of these scores.

Definitions and further discussion of these terms are available from the following resources;

·       National Research Council (US) Committee on Recognition and Alleviation of Distress in Laboratory Animals. Recognition and Alleviation of Distress in Laboratory Animals. Washington (DC): National Academies Press (US); 2008. PMID: 20669418.https://nap.nationalacademies.org/catalog/11931/recognition-and-alleviation-of-distress-in-laboratory-animals

·       Australian code for the care and use of animals for scientific purposes https://www.nhmrc.gov.au/about-us/publications/australian-code-care-and-use-animals-scientific-purposes

Typographical errors

Line 48 – This sentence is incomplete ‘The shrimp farming production chain is considered one of the most controversial in aquaculture, one of the most agri-food systems for animal protein production in the world’

Line 50 – ‘The evolution of shrimp farming practises currently taking place on several fronts.’ Write ‘The evolution of shrimp farming practises is currently taking place on several fronts.

Line 80 ‘sentient’ should be ‘sentience’

Line 84 ‘From a first perspective’ should be ‘From the first perspective’

Line 85 It is not clear what is meant by ‘the creation of positive conditions for that suffering’

Line 222 ‘More specifically were visited’ should be ‘More specifically we visited’

Line 230 ‘they are presented in tables 2 to 15’ should be ‘they are presented in tables 2 to 14’ There is no table 15.

Line 410 This sentence is incomplete ‘Finally, clinical reflexes during stunning at slaughter are 410 somewhat technical and meticulous analysis but aimed at evaluating one of the most 411 critical points in this rearing phase.’

Line 420 ‘Sustainability of production, the environment and animal welfare is a critical issue in’ should be ‘Sustainability of production, the environment and animal welfare are critical issues in’

Line 427 ‘Concerning breeders, the principal reported risks to relate to the removal of the’ should be ‘Concerning breeders, the principal reported risks relate to the removal of the’

Line 483 ‘In the case of shrimp, invasive interventions are unlikely to affect their welfare seriously, so we try to minimise them in our protocols.’ Should be ‘In the case of shrimp, invasive interventions are likely to affect their welfare seriously, so we tried to minimise them in our protocols.

Line 539 Is PL22 correct? i.e. are there 22 post-larval stages?

Line 544 ‘such as dissolved oxygen solubility’ should read ‘such as dissolved oxygen’ or ‘such as solubility of oxygen’

Line 574 ‘all these practises on the’ should be ‘all these practices on the’

Line 582 A space is missing between the sentences on this line

Line 595 ‘efficiency of management practises applied’ should be ‘efficiency of management practices applied’

Reviewer 3 Report

Overall, the manuscript is well-prepared. However, minor revision is required as per the

comments provided in relevant sections with specific line number.

Introduction:

ï‚· Line no 50: in the world (see [1–5]). Delete (see).

ï‚· Line no 99-100: The brains …….. from humans. Include reference.

Results:

ï‚· Line no 234-235: Mention the age/size/weight of P. vannamei breeder.

ï‚· Line no 295-296: Explain different stages involved between larvae to post larvae, like

how much stages, time of changes etc.

ï‚· Line no 346: Which size is considered as post larvae?

Discussion:

ï‚· Line no 567-569: The ecological and biological …….. for shrimp farming. Include reference.
